# Single-cell RNA sequencing analysis of shrimp immune cells identifies macrophage-like phagocytes

**Peng Yang[1], Yaohui Chen[2], Zhiqi Huang[2], Huidan Xia[2], Ling Cheng[3], Hao Wu[3], Yueling Zhang[1,4,5], Fan Wang[1,2,4]***

[1]Institute of Marine Sciences, Guangdong Provincial Key Laboratory of Marine Biotechnology, Shantou University, Shantou, China; [2]Department of Biology, College of Science, Shantou University, Shantou, China; [3]Guangzhou Genedenovo Biotechnology Company Limited, Guangzhou, China; [4]Shantou University-Universiti Malaysia Terengganu Joint Shellfish Research Laboratory, Shantou University, Shantou, China; [5]Southern Marine Science and Engineering Guangdong Laboratory, Guangzhou, China

**Abstract** Despite the importance of innate immunity in invertebrates, the diversity and function of innate immune cells in invertebrates are largely unknown. Using single-cell RNA-seq, we identified prohemocytes, monocytic hemocytes, and granulocytes as the three major cell-types in the white shrimp hemolymph. Our results identified a novel macrophage-like subset called monocytic hemocytes 2 (MH2) defined by the expression of certain marker genes, including *Nlrp3* and *Casp1*. This subtype of shrimp hemocytes is phagocytic and expresses markers that indicate some conservation with mammalian macrophages. Combined, our work resolves the heterogenicity of hemocytes in a very economically important aquatic species and identifies a novel innate immune cell subset that is likely a critical player in the immune responses of shrimp to threatening infectious diseases affecting this industry.

*For correspondence:
wangfan@stu.edu.cn

## Editor's evaluation

This study provides a single cell transcriptomic atlas of shrimp hemocytes and identifies a subset of myeloid cells with markers that resemble mammalian macrophages. These novel phagocytic macrophage subset may be the target of future studies in diseased shrimp.

## Introduction

Compared with vertebrates, invertebrates do not have B- and T-cell-based adaptive immunity, which makes them primarily reliant on innate immunity as defense against various pathogens (*Little et al., 2005*). Although invertebrates have developed diverse forms of innate immunity in order to adapt to various environmental challenges, the innate immune cells themselves contribute significantly because they have 'trained immunity' (*Lanz-Mendoza and Contreras-Garduño, 2022*). However, the evolution of innate immune systems has diverged into many branches in the metazoan tree of life, making invertebrate immune cell-typing extremely complex. In contrast, the cells of the vertebrate immune system are conserved from chordates (*Rosental et al., 2018*). Recent advances in single-cell sequencing technology have shed light on this issue. For example, recent research in mosquito cellular immunity revealed that mosquito hemocytes are of four major types (prohemocytes, granulocytes, oenocytoids, and megacytes) (*Kwon et al., 2021*; *Raddi et al., 2020*). Hemocytes from another invertebrate

model — *Drosophila melanogaster* — can be divided into eight subgroups, including crystal cells, lamellocytes, unspecified plasmatocytes, proliferative plasmatocytes, PSC-like hemocytes, antimicrobial plasmatocytes, phagocytic plasmatocytes, and secretory plasmatocytes (*Cattenoz et al., 2021*; *Cho et al., 2020*; *Li et al., 2022*; *Tattikota et al., 2020*). These classifications cover the major innate immune cell functions, including proliferation, reactive oxygen species (ROS) generation, phagocytosis, and effector secretion, suggesting some shared similarities in innate immunity between invertebrates and vertebrates.

To explore these analogies, we used the marine invertebrate *Penaeus vannamei* to characterize immune cell subsets using single-cell RNA sequencing (scRNA-seq). This species was selected because it is a popular mariculture species due to being fast growing and delectable (*Zhang et al., 2019*). Additionally, this invertebrate has an open circulatory system filled with hemolymph, which consists of plasma and hemocytes (*Lin and Söderhäll, 2011*). The shrimp hemolymph bears high similarity to that of the vertebrate peripheral blood in terms of functions such as transportation of nutrients and metabolic waste, maintenance of acid–base equilibrium, defense against various invading pathogens, and hemostatic effect (*McNamara and Faria, 2012*; *Tassanakajon et al., 2018*). Moreover, the circulating plasma contains more than 400 proteins, many of which are vertebrate homologs (*Luo et al., 2022*; *Tao et al., 2019*). The circulating hemocytes comprise proliferating cells, phagocytic cells, and effector secreting cells (*Lin and Söderhäll, 2011*), which are functionally similar to vertebrate peripheral myeloid cells. Among these cells, the phagocytes have been identified in all metazoans (*Musser et al., 2021*), although they appear to be different in invertebrates and vertebrates. Some recently performed functional assays suggest that the bacterial engulfment in phagocytes may be conserved from invertebrates to vertebrates (*Kokhanyuk et al., 2021*). In this study, we attempted to analyze shrimp hemocytes via single-cell sequencing and redefined their classification according to their functional marker gene distribution. We also attempted to understand evolutionary phagocyte development along the metazoan tree of life.

## Results
### Major cell-types among the circulating hemocytes in shrimp

Previously, we identified a lipopolysaccharide (LPS)-induced shrimp plasma protein CREG and noticed that recombinant CREG (rCREG) was more effective than recombinant EGFP (rEGFP) in activating the shrimp hemocytes (Huang, Yang, & Wang, 2021). Based on this observation, we performed scRNA-seq for rCREG-treated shrimp hemocytes to further explore their function. To maximize the collection of circulating hemocytes from shrimp, we applied iodixanol gradient centrifugation to concentrate the hemocytes for the Gel Bead-In-Emulsion (GEM) preparation (*Tattikota et al., 2020*; *Figure 1A*). A total of 34,693 cells including control (12544), rEGFP- (12640), and rCREG-treated (9509) cells were retained for further analyses, and these cells exhibited a median of 5656, 6837, and 7916 transcripts and 1089.5, 1245, and 1364 genes per cell, respectively (*Figure 1—figure supplement 1*). The rCREG-treated samples had the highest unique molecular identifiers (UMIs) and detected genes compared with that of the other two groups. This observation is consistent with our previous conclusion that CREG is a hemocyte activation factor (*Huang et al., 2021*).

To further define the major cell types of circulating shrimp hemocytes, we combined all 34,693 cells from different treatments and applied canonical correlation analysis (*Stuart et al., 2019*) to perform batch correction. We then aggregated the cell clusters and identified five major groups of isolated hemocytes, including prohemocytes (PHs) (6838, 19.7%), granulocytes (GHs) (13871, 40%), monocytic hemocytes (MHs) (11112, 32%), transitional cells (TCs) (2090, 6%), and germ-like cells (GCs) (782, 2.3%), which were annotated according to their potential functions implied by the marker genes (*Figure 1B*). Granulocytes are characterized by their ProPO system (*Sun et al., 2020*). Here, we found that prophenoloxidase activating factors 1 and 2 (*PPAF1* and *PPAF2*) were highly expressed in this population. In addition to ProPO system genes, secreted proteins, including crustin-like protein (*CRUL*), penaeidin 3 a.1 (*PEN-3*), and crustacean hematopoietic factor-like protein (*CHF*), were also highly expressed in this population (*Figure 1C*, *Figure 1—figure supplement 2A*). The MHs were named thus because they shared some critical genes with mammalian monocytes. For example, NOD-like receptor protein 3 (*Nlrp3*) is a key component of the inflammasome and is highly expressed in monocytes and macrophages for processing of IL-1ß (*He et al., 2016*). Lysosome (*Lyz1*), another

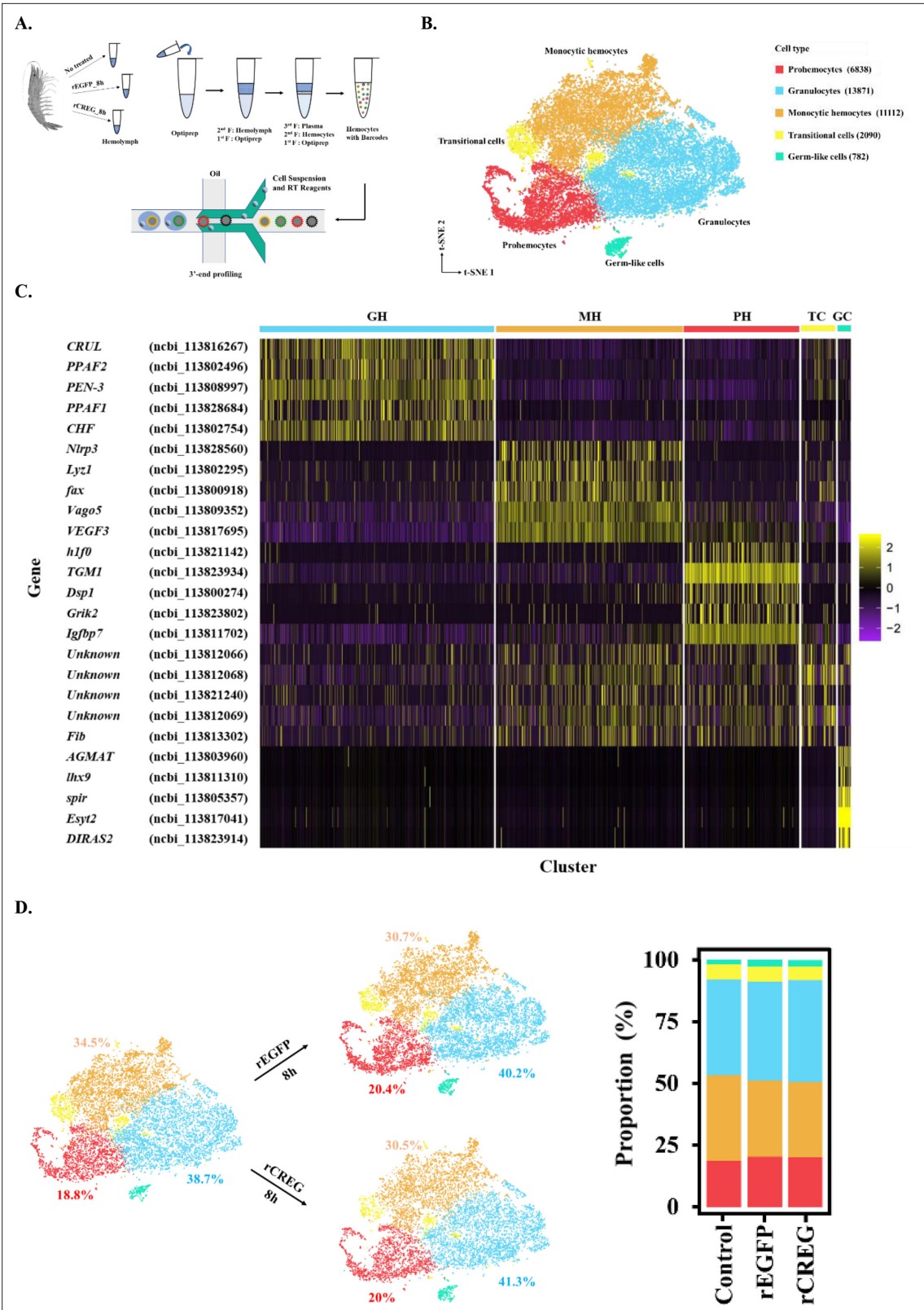

**Figure 1.** Major cell types identified in shrimp hemolymph. (**A**) A schematic workflow of sample preparation. The hemocytes were collected from non-treated, rEGFP-treated, and rCREG-treated shrimps (n=15 for each treatment) and subjected to iodixanol gradient centrifugation and single-cell RNA sequencing (ScRNA-seq) using 10 X Genomics. (**B**) A t-SNE plot showing five major cell types identified in scRNA-seq dataset (n=34,693 in total; Control, 12544; rEGFP treated, 12640; rCREG treated, 9509 cells). The count of each cell type is indicated in parentheses. (**C**) A heatmap showing five

*Figure 1 continued on next page*

Figure 1 continued

representative marker genes for each major cluster. The gene name and its NCBI GeneID is listed (left) and its expression level in each cell is shown with different colors (right). (**D**) Two-dimensional projections, and proportions of the cell types for each treatment. Proportions of prohemocytes (red), monocytic hemocytes (brown), and granulocytes (blue) are indicated (left). Proportions of all five major cell types in each treatment are indicated (right).

The online version of this article includes the following figure supplement(s) for figure 1:

**Figure supplement 1.** Overall quality of single-cell transcriptomic data.

**Figure supplement 2.** Distribution of the marker genes in major cell types.

canonical antibacterial enzyme, is a well-known macrophage-secreting hydrolase (**Short et al., 1996**). *Vago5* encodes an IFN-like antiviral cytokine and plays a role in the anti-white spot syndrome virus (WSSV) resistance (C. Li, Yang, Hong, Zhao, & Wang, 2021) (**Figure 1C**, **Figure 1—figure supplement 2B**). Generally, these cells secrete antibacterial and antiviral effectors. In PHs, histone1 (*h1f0*) is the key component of heterochromatin assembly, and its high expression is associated with reduced gene expression and cell stemness properties (**Pan and Fan, 2016**). In a previous study, hemocyte trans-glutaminase (*TGM1*) was identified as an immature hemocyte marker (**Koiwai et al., 2021**). The gene *Igfbp7* has been shown to promote hemocyte proliferation in the small abalone *Haliotis diversicolor* (**Wang et al., 2015**; **Figure 1C**, **Figure 1—figure supplement 2C**). TCs were difficult to characterize due to the lack of exclusively expressed genes. Hence, we identify the top five significantly upregulated genes and analyzed the distribution of the top three (**Figure 1C**, **Figure 1—figure supplement 2D**); we found that this group of cells had no significant marker genes, thus the name. We found some reproduction-related genes in the GC group. For example, *lhx9* encodes a key transcription factor in gonadal development (**Balasubramanian et al., 2014**), and *spir* localization is critical for mouse oocyte asymmetric division (**Jo et al., 2019**; **Figure 1C**, **Figure 1—figure supplement 2E**). Hence, we annotated this group of cells as GCs.

To further determine whether CREG is a differentiation factor for shrimp hemocytes, we examined the ratio of the five annotated major cell types in different treatments. Recombinant protein injection slightly increased the proportion of GHs and PHs and decreased the ratio of MHs (**Figure 1D**). However, there were no significant differences between the rEGFP and rCREG treatments, which suggests that CREG is probably an activation factor rather than a differentiation factor for shrimp hemocytes.

## Subtyping of shrimp immune cell clusters and construction of their differentiation trajectory

While classifying the immune cells in shrimp hemolymph, the TC group that lacks gene markers and GC group that is not a typical immune cell are not further explored in this study (**Supplementary file 1**, **Supplementary file 2**). To further trace the immune cell lineages in shrimp hemolymph, we subtyped the remaining three major classes of cells—PHs, MHs, and GHs; each major class type was divided into two subtypes and labelled PH1 (1577, 4.5%) and PH2 (5261, 15.2%), MH1 (10463, 30.2%) and MH2 (649, 1.9%), and GH1 (10353, 29.8%) and GH2 (3518, 10.1%), respectively (**Figure 2A**). We identified some unique marker genes located at the edge of the t-SNE map for the subpopulations PH1, GH2, and MH2 (**Figure 2A**, **Figure 2—figure supplement 1**, **Supplementary file 3**, **Supplementary file 4** and **Supplementary file 5**), but could not identify exclusive marker genes for the subpopulations PH2, GH1, and MH1, which constituted the main body of the t-SNE map. The marker genes for PH2, GH1, and MH1 were also highly expressed in PH1, GH2, and MH2, respectively (**Figure 2B**, **Supplementary file 6**, **Supplementary file 7** and **Supplementary file 8**). Thus, these six subtypes of hemocytes might have lineage differentiation relationships. To explore this, we performed cell cycle analyses of the six subtypes of hemocytes. PH1 was characterized by high expression of all marker genes for the G1, G2, and M stages (**Figure 2C**). This observation was consistent with a previous report that approximately 2–5% of circulating hemocytes were proliferating hemocytes and could be labelled with BrdU (**Sun et al., 2013**). Thus, we set PH1 as the initiating cell and applied a Monocle to construct differentiation trajectories for PHs, GHs, and MHs. Two major branches—MH lineage and GH lineage—were identified to have differentiated from one common PH (**Figure 2D–E**). This is similar to the development of human myeloid cells, in which granulocyte–monocyte progenitor (GMP) differentiates into monocyte and granulocyte (**Bassler et al., 2019**). To further compare innate immune cell differentiation

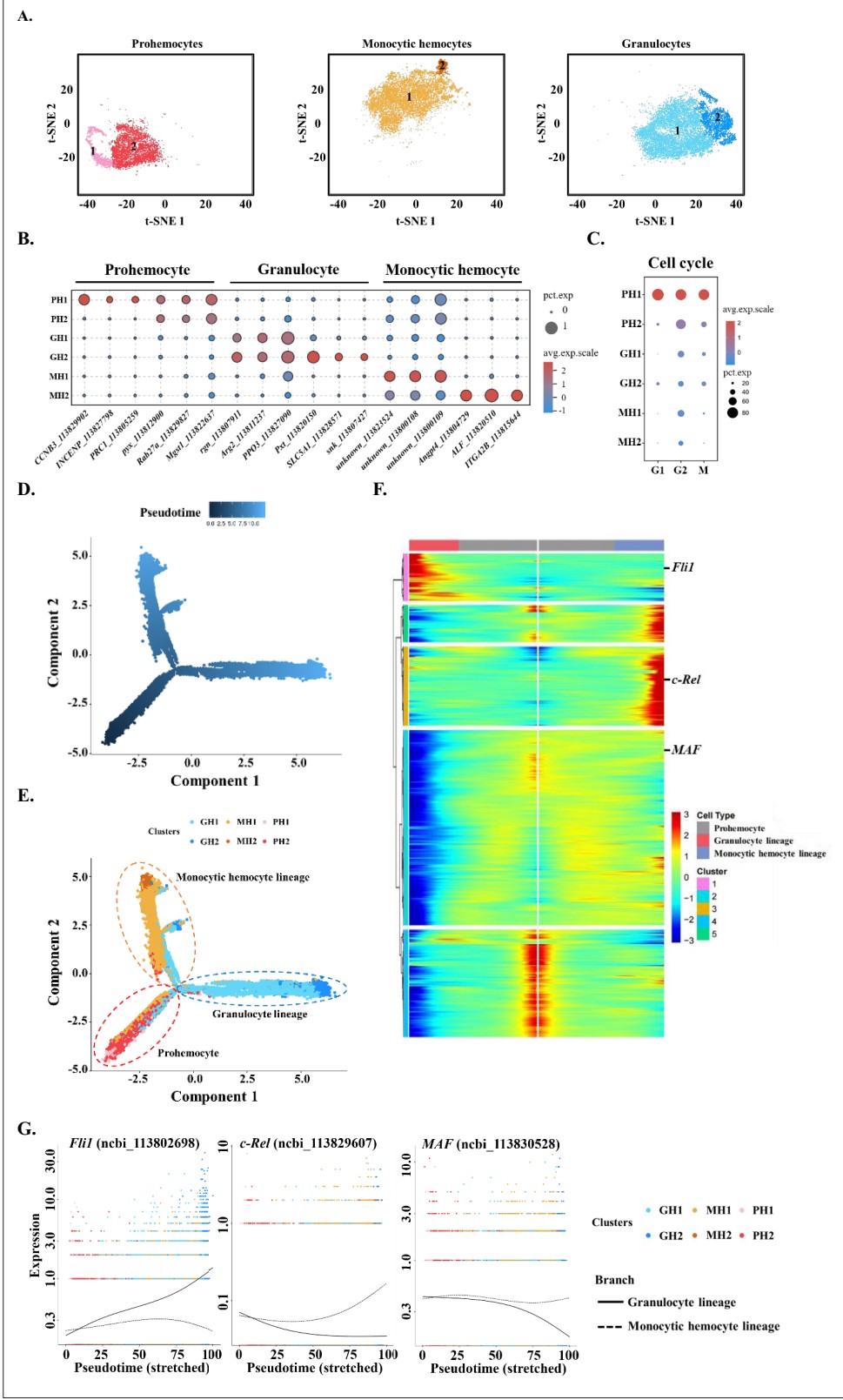

**Figure 2.** Subclustering and pseudotime trajectory analyses of three major hemocyte types in shrimp hemolymph.
(**A**) Subclusters of hemocytes–prohemocytes (PH), monocytic hemocytes (MH), granulocytes (GH) are projected
onto two-dimensional t-SNE plots. The numbers in the plots represent the subcluster number. (**B**) Dot plot
showing corresponding expression of cluster marker genes. The color indicates mean expression and dot size

*Figure 2 continued on next page*

*Figure 2 continued*

represents the percentage of cells within the cluster expressing the marker. Last nine digits of each marker gene are the NCBI GeneID. (**C**) Expression of cell-cycle regulating genes in the six subtypes. Dot color indicates average expression levels and dot size displays the average percentage of cells with cell cycle controlling genes (*Cdk1*(ncbi_113818305), *CycD*(ncbi_113814652), and *CycE*(ncbi_113822658) for G1; *stg*(ncbi_113800052), *CycA*(ncbi_113821735), and *CycB*(ncbi_113803283) for G2; *polo*(ncbi_113805901), *aurB*(ncbi_113827838), and *birc5*(ncbi_113828653) for M) in each subcluster. (**D**) A differentiation trajectory of PH, GH, and MH subpopulation using Monocle2 (n=31821). (**E**) Differentiation trajectory reconstruction with 6 subclusters. PH lineage, GH lineage, and MH lineage were labelled with red, blue, and brown circles respectively. (**F**) A heatmap showing differentially expressed gene dynamics during hemocyte differentiation process. (**G**) Spline plots showing the expression dynamics of *Fli1*, *c-Rel*, and *MAF*. Imaginary line, monocytic hemocyte lineage; Full line, granulocyte lineage.

The online version of this article includes the following figure supplement(s) for figure 2:

**Figure supplement 1.** Distribution of the marker genes for PH1, MH2 and GH2.

---

between shrimp and humans, we screened shrimp homologs of human myeloid differentiation-related transcription factors (TFs) because TFs are the key regulators of cell fate determination (*Friedman, 2002*). In total, 3790 differentially expressed genes among different branches were identified and are shown as a specific heatmap on which three shrimp homologs of human TFs were labeled (*Figure 2F*, *Supplementary file 9*). *Fli1* is specifically expressed in the granulocyte lineage, which is consistent with previous observations that *Fli1* deletion decreases granulocytic cell number in mice (*Starck et al., 2010*). *MAF* and *c-Rel* were highly expressed in the MH lineage (*Figure 2G*). *c-Rel* is a key TF in the NF-κB pathway and plays important roles in monocyte differentiation (*Li et al., 2020*). *MAF* is a bZip TF that could induce monocytic differentiation (*Kelly et al., 2000*). In general, our data suggest that some myeloid regulators may be conserved between shrimp and humans.

## Identification of a macrophage-like phagocytic cell population in shrimp hemolymph

Next, we analyzed the similarities that MH2 might be sharing with terminally differentiated monocyte-like macrophages or dendritic cells. Recently, the Human Cell Atlas mapped the expression of most genes across major human cell types (*Karlsson et al., 2021*). We compared MH2 marker genes with that in the human database and found human homologs for nine MH2 marker genes including chitotriosidase (*CHIT1*), lysozyme (*Lyz1*), lipase (*LIPF*), legumain (*LGMN*), Nlrp3, alpha-N-acetylgalactosaminidase (*NAGA*), zinc finger E-box-binding homeobox 1(*zfh1*), caspase1 (*Casp1*), and NPC intracellular cholesterol transporter 2 (*NPC2*). These genes were specifically expressed in human macrophages, including in some tissue-specific macrophages such as Kupffer cells and Hofbauer cells (*Figure 3A*, *Figure 3—figure supplement 1*, *Figure 1—figure supplement 2B*; *Karlsson et al., 2021*). This suggests that MH2 might be the invertebrate homolog of human macrophages, and various tissue-specific macrophages could have evolved from a common primitive cell type.

To further prove this hypothesis, we labeled phagocytes via injection of fluorescein isothiocyanate-conjugated *Vibrio parahaemolyticus* (FITC-VP). The hemocytes which engulfed FITC-VP were isolated using a cell sorter and labelled as phagocytic hemocytes (R1) (fluorescence intensity >2 × 10$^3$). The hemocytes with low fluorescence (<10$^3$) were labelled control hemocytes (R2) (*Figure 3B*). To characterize these phagocytes, we observed them using confocal microscopy. These cells had round nuclei with a highly vacuolated cytoplasm similar to that in vertebrate macrophages (*Figure 3C*). To further compare these cells with vertebrate macrophage, we performed a phagocytosis inhibition assay using an actin polymerization inhibitor—cytochalasin D (*Kokhanyuk et al., 2021*)—that effectively suppressed the shrimp hemocyte phagocytosis rate (*Figure 3D*). To characterize the R1 cells, we quantified *CHIT1*, *Lyz1*, and *NAGA* expression in R1 and R2 using qPCR. The results indicated that these three genes were expressed at higher levels in R1 than in R2 (*Figure 3E*). In addition, we examined LYZ1, NAGA, and NLRP3 using immunoblotting and found that these three proteins were expressed at significantly higher levels in R1 than in R2 (*Figure 3F*). Thus, our results indicated that phagocytic cells in shrimp hemolymph specifically expressed MH2 marker genes. Our data suggest that MH2 may be an invertebrate homolog of human macrophages.

**A.**

| | MH2 marker genes | | | Human homologs of MH2 marker genes | | | | Sequence alignment | |
|---|---|---|---|---|---|---|---|---|---|
| Target Cluster | Gene ID | Gene Name | MH2 specificity (Log2FC) | Ensembl 92 ID | NCBI Gene ID | Specificity score | Elevated cell types | Identity % | Cover % |
| MH2 | ncbi_113822241 | CHIT1 | 5.608553996 | ENSG00000133063 | 1118 | 44.23458957 | Hofbauer cells | 46 | 55 |
| MH2 | ncbi_113802295 | Lyz1 | 4.743882459 | ENSG00000090382 | 4069 | 15.86859173 | Kupffer cells; Macrophages; Pancreatic endocrine cells | 45 | 55 |
| MH2 | ncbi_113823443 | LIPF | 4.720324613 | ENSG00000107798 | 3988 | 7.429198963 | Hofbauer cells; Kupffer cells; Macrophages | 37 | 98 |
| MH2 | ncbi_113827868 | LGMN | 4.337645965 | ENSG00000100600 | 5641 | 15.339588 | Hofbauer cells; Ito cells; Syncytiotrophoblasts | 53 | 79 |
| MH2 | ncbi_113828560 | Nlrp3 | 4.182391747 | ENSG00000162711 | 114548 | 11.12063946 | Hofbauer cells; Kupffer cells; Macrophages | 25 | 26 |
| MH2 | ncbi_113810366 | NAGA | 3.983237214 | ENSG00000198951 | 4668 | 5.088576154 | Hofbauer cells | 52 | 92 |
| MH2 | ncbi_113824903 | zfh1 | 3.501526253 | ENSG00000169554 | 6935 | 10.34917618 | Horizontal cells; Kupffer cells; Macrophages | 77 | 22 |
| MH2 | ncbi_113800234 | Casp1 | 3.202709678 | ENSG00000137752 | 841 | 6.77037223 | Hofbauer cells; Kupffer cells; Macrophages | 26 | 51 |
| MH2 | ncbi_113806913 | NPC2 | 3.188555286 | ENSG00000119655 | 10577 | 6.899977629 | Alveolar cells type 2; Hofbauer cells; Kupffer cells | 37 | 53 |

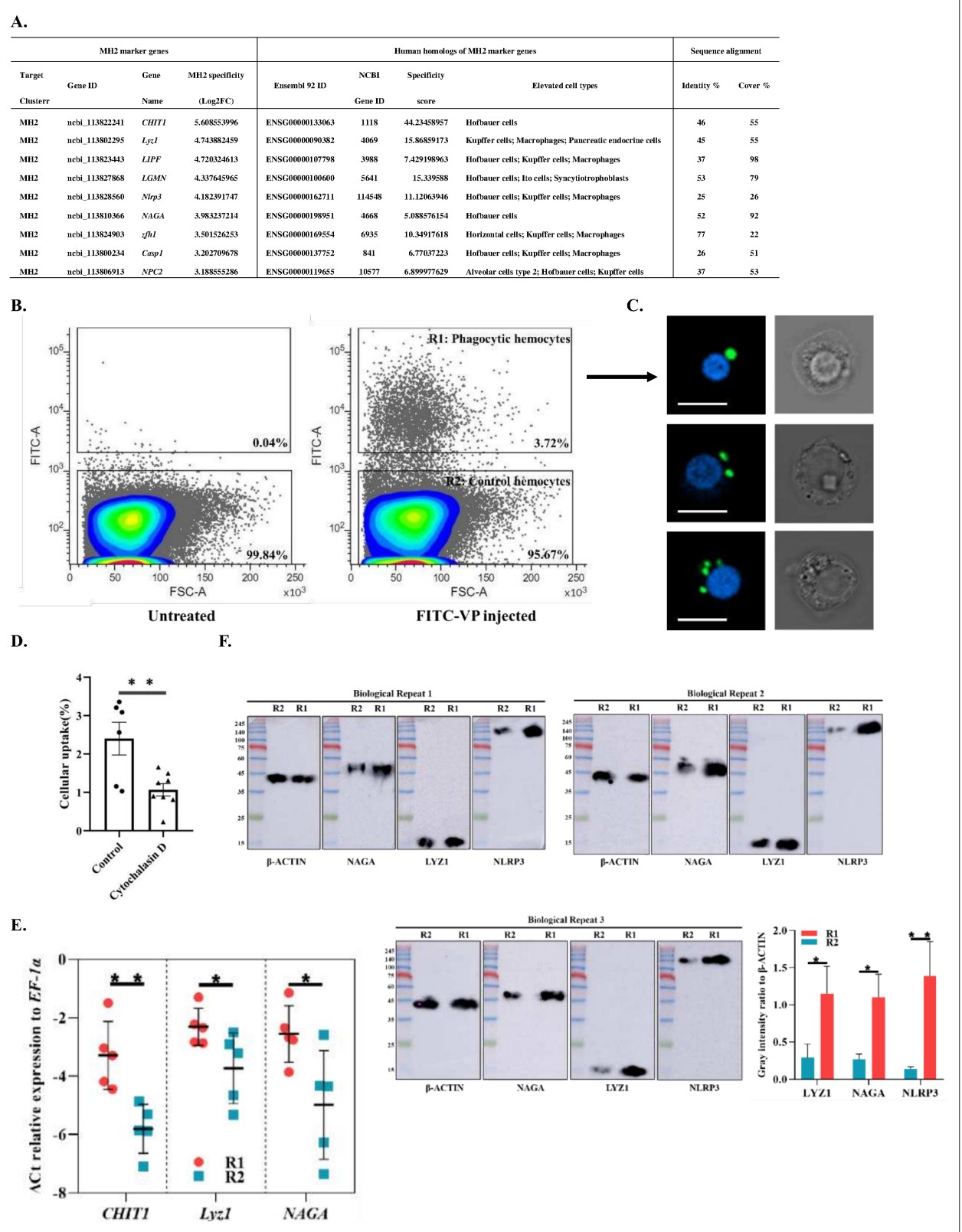

**Figure 3.** Identification of MH2 as macrophage-like phagocytic cells. (**A**) Comparison between MH2 and human macrophage marker genes. (**B**) A representative contour plot of shrimp hemocytes against FITC-VP. Threshold intensity (FITC-A) was set to <10$^3$ representing control hemocytes (R2), and >2 × 10$^3$ representing phagocytic hemocytes (R1). R1 and R2 were sorted based on the forward scatter (FSC) and fluorescence intensity (FITC) two-dimensional space. (**C**) Confocal microscopy of sorted hemocytes (R1) with ingested FITC-labelled *Vibrio Parahemolyticus*. Green, ingested *Vibrio*

*Figure 3 continued on next page*

*Figure 3 continued*

*Parahemolyticus*; Blue, nuclei. Scale bar: 10 µM. (**D**) Efficiency of the phagocytosis inhibitor on the *Vibrio Parahemolyticus* uptake of shrimp hemocytes. The results are presented as mean ± SD of 6–8 replicates. Asterisks denote statistical significance (**p=0.007) between the control and different treatments. (**E**) Differential gene expression analysis (*CHIT1* (**p=0.004), *Lyz1* (*p=0.049), and *NAGA* (*p=0.032)) between R1 and R2 sorted using FACS and analyzed using qPCR. (**F**) Differential protein expression analysis (NAGA, LYZ1, and NLRP3) between R1 and R2 sorted using FACS. The immunoblot signals were quantified with ImageJ. The relative immunoblot signal intensities of NAGA (*p=0.011), LYZ1 (*p=0.022), and NLRP3 (**p=0.009) compared with that of ß-actin were recorded with bar chart. Both qPCR and immunoblot data were analyzed using the student *t* test.

The online version of this article includes the following source data and figure supplement(s) for figure 3:

**Source data 1.** Flow cytometry source data for *Figure 3D*.

**Source data 2.** Raw qPCR data for *Figure 3E*.

**Source data 3.** Raw and labeled western blot source images for *Figure 3F*.

**Figure supplement 1.** Distribution of MH2 marker genes, which are conserved with that of human macrophages.

## Comparison between hyalinocytes, semi-granulocytes, and granulocytes and their classifications in this study

Next, we compared our classification with the traditional classification. Previously, shrimp hemocytes have been divided into three major types: hyalinocytes, semi-granulocytes, and granulocytes based on morphological criteria and functional properties (*Söderhäll, 2016*). Recently, these three major types were separated using cell sorting or Percoll density gradient centrifugation and their marker genes were identified and validated using qPCR (*Sun et al., 2020*; *Yang et al., 2015*). Here, we analyzed the distribution of previous published marker genes: for hyalinocytes—lysosome membrane protein2 (*LIMP2*, ncbi_113826216), tubulin beta chain (*TUBB4B*, ncbi_113826677), dipeptidyl peptidase 1 (*CTSC*, ncbi_113824311), transglutaminase 1 (*TGM1*, ncbi_113823934) (*Figure 4A*, *Figure 1—figure supplement 2C*); for semi-granulocytes—beta-arrestin-1 (*ARRB1*, ncbi_113804686), ADP-ribosylation factor 6 (*ARF6*, ncbi_113820333), lysozyme (*Lyz1*, ncbi_113802295), Penaeid-3a (*PEN-3*, ncbi_113808997) (*Figure 4B*, *Figure 1—figure supplement 2A and B*); and for granulocytes—clone ZAP 18 putative antimicrobial peptide (*CRU*, ncbi_113801825), phenoloxidase-activating factor 3 (*PPAF3*, ncbi_113800184), phenoloxidase 3-like (*PPO3*, ncbi_113827090), peroxinectin (*Pxt*, ncbi_113820150) (*Figure 4C*, *Figure 2—figure supplement 1C*). Hyalinocyte marker genes were highly expressed in PH1, PH2, MH1, and MH2 groups (*Figure 4A and D*). Semi-granulocyte marker genes were highly expressed in GH2 and MH2 groups (*Figure 4B and D*). These data are consistent with previous observations that hyalinocytes contain both proliferating progenitors and phagocytic cells (*Söderhäll, 2016*). It also explained the observation of phagocytic activities in semi-granulocytes in some studies (*Sun et al., 2020*). The granulocyte marker genes were consistent with our observations and were highly expressed in GH2 (*Figure 4C–D*), which indicates that granulocytes are indeed the largest cell-type with internal condensed granules (*Söderhäll, 2016*).

## Discussion

Innate immune cells play an important role in the adaptation of animals to complex and volatile environments. Their ability for fast response protects animals from various pathogenic invasions. However, invertebrates have experienced a long evolution in a diversified environment, which has led to the fact that invertebrate immunity is extremely complex and immune cell typing in various invertebrates seems quite different (*Supplementary file 10*). Fortunately, proteins seem to have evolved at a much slower rate (*Jayaraman et al., 2022*), and cell-specific functional proteins are therefore the key to define cell subsets. Thus, in this study, we compared shrimp immune cell marker genes with their human homologs to identify evolutionary traces of innate immune cells between invertebrates and vertebrates (*Supplementary file 11*). Our data revealed macrophage-like phagocytes in shrimp hemolymph. This group of cells exhibited phagocytic activity. Additionally, *Nlrp3* and *Casp1*, two well-known mammalian macrophage inflammasome components were identified in these cells, suggesting that inflammasome-mediated anti-pathogenic processes might exist in invertebrate innate immunity and play a role in microbial restriction (*Wang et al., 2021*). NAGA, another enzyme found in macrophages, could inhibit macrophage activation via deglycosylation of macrophage activators (*Saburi et al., 2017*). The existence of NAGA in shrimp phagocytes implies that this subset of cells

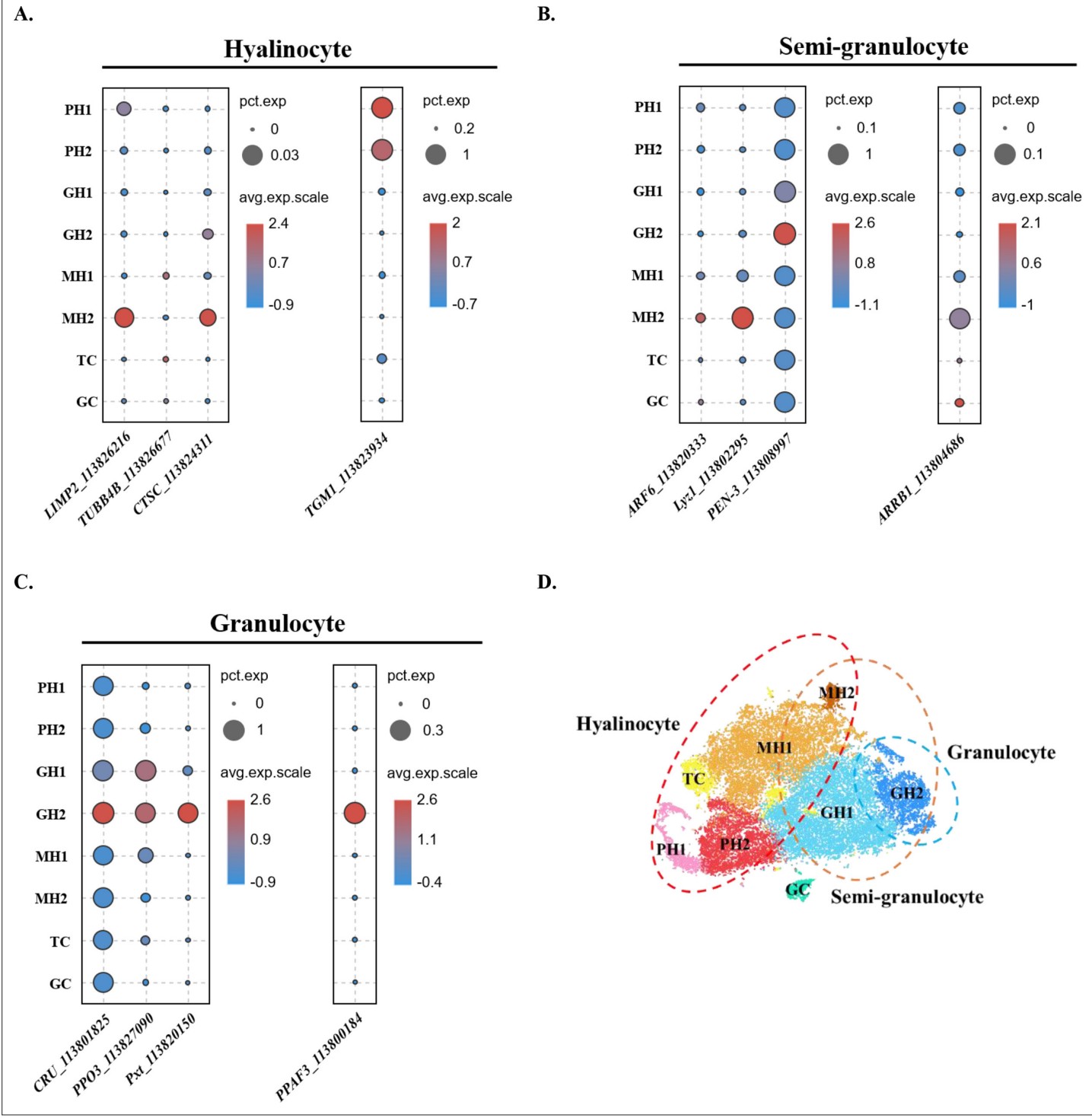

**Figure 4.** Comparison between the traditional classification and the classification in this study. (**A**) Dot plot showing corresponding expression of previously reported hyalinocyte marker genes in eight subclusters. (**B**) Dot plot showing corresponding expression of previously reported semi-granulocyte marker genes in eight subclusters. (**C**) Dot plot showing corresponding expression of previously reported granulocyte marker genes in eight subclusters. The color indicates mean expression, and dot size represents the percentage of cells within the cluster expressing the marker. (**D**) A proposed model for comparison between two classifications. The hyalinocyte, semi-granulocyte, and granulocyte were labelled on the t-SNE map with red, brown, and blue circles, respectively.

might share similar negative regulatory mechanisms recently partially uncovered in vertebrate macrophages (*Luo et al., 2022*). In addition, VEGF3, a well-known angiogenic factor, was also identified in this subtype (*Eswarappa and Fox, 2015*). Shrimp have an open circulation system with partial blood vessels; whether this factor plays a role in wound healing needs to be answered by future studies (*Söderhäll, 2016*).

Phagocytic ability is one of the fundamental functions in an organism. Unicellular organisms employ phagocytosis for this purpose. Cells in multicellular organisms have functional specializations that increase their adaptability. Thus, the phagocytic ability of metazoans is limited to certain cells. The ratio of phagocytic cells in the different species varies. For example, in the primitive oyster *Crassostrea gigas*, approximately 40–60% of hemocytes engulf pathogens (*Sun et al., 2021*). In fish, most immune cells, including monocytes, granulocytes, B cells, and red blood cells, possess phagocytic activity against invading pathogens (*Heimroth et al., 2021*; *Li et al., 2006*; *Xu et al., 2021*). In humans, most myeloid cells, including monocytes, macrophages, dendritic cells, and neutrophils, engulf pathogens (*Bassler et al., 2019*). Shrimps contain fewer phagocytes than do other species (*Alenton et al., 2019*; *Huang et al., 2021*; *Li et al., 2021*). Our data suggest that at least a part of the phagocytes in the shrimp hemolymph originated from a macrophage-like subset. Moreover, vertebrate macrophages include large populations of cells that reside in different tissues with diversified roles. Whether invertebrate macrophages have similar tissue-specific functions is yet unknown and future studied are needed to elucidate the evolution of these helper cells in maintaining homeostasis.

Over the past decades, crustacean hemocytes have been classified into hyalinocytes, semigranulocytes, and granulocytes based on their morphology and function (*Söderhäll, 2016*). However, morphology-based classification has caused several problems. For example, crustacean hyalinocytes are generally regarded as small phagocytes with few granules (*Lin and Söderhäll, 2011*), but some researchers believe that these cells may be immature or prematurely released prohemocytes of the semi-granulocyte or granulocyte lineage (*van de Braak et al., 2002*). In this study, our data clearly indicated that the small cells though similar in morphology include both prohemocytes and phagocytic hemocytes, and these two subtypes of cells have different marker genes and varied functional roles in the shrimp hemolymph. In addition, semi-granulocytes were considered a major component of circulating hemocytes that were involved in both melanization and phagocytosis. This conclusion cannot be accurate because less than 20% of circulating hemocytes are phagocytic cells, whereas approximately 65% of total hemocytes are considered to be semi-granulocytes (*Alenton et al., 2019*; *Lin and Söderhäll, 2011*). Our results indicated that the cells sorted as semi-granulocytes contain both monocytic hemocytes and cells of granulocyte lineage.

In this study, we selected *Penaeus vannamei* as a crustacean model and proposed a novel crustacean hemocyte classification system. A new functional monocytic hemocyte lineage for circulating shrimp hemocytes was identified. The terminally differentiated cells of this lineage share several functional genes with mammalian macrophages, which suggests that this monocytic hemocyte lineage might be an invertebrate evolutionary homolog of the vertebrate monocyte lineage. Although crustaceans contain diverse species with different morphologies and evolutionary histories, our data, to some extent, provide a different interpretation for the current crustacean immune cell subtyping. This classification is far from complete but might provide insights about crustacean cellular immunity in the future. Moreover, this study is at an early stage of investigating cellular immunity in shrimp. For example, we could not systematically validate our conclusion at this moment due to the lack of specific monoclonal antibodies to label the three proposed major immune cell subtypes. Some fundamental questions remain unresolved, such as: can shrimp macrophage-like phagocytes infiltrate various tissues and where do the circulating shrimp prohemocytes originate? In future studies, we plan to find answers to such questions to improve our understanding of this important mariculture species and find solutions to serious shrimp diseases that have caused tremendous economic losses worldwide.

## Materials and methods

### Key resources table

| Reagent type (species) or resource | Designation | Source or reference | Identifiers | Additional information |
|---|---|---|---|---|
| Antibody | Anti-ß-ACTIN (Rabbit monoclonal) | Beyotime | Cat#AF5003 | WB (1:200) |
| Antibody | Anti-NAGA (Rabbit polyclonal) | SinoBiological | Cat#13686-T24 | WB (1:200) |
| Antibody | Anti-LYZ1 (Rabbit polyclonal) | Bioss Antibodies | Cat#bs-0816R | WB (1:200) |
| Antibody | Anti-NLRP3 (Rabbit polyclonal) | GenScript | | polypeptide (aa29-42) WB (1:200) |
| Strain, strain background (*Vibrio parahaemolyticus*) | FITC-VP | Shantou University | | $2 \times 10^6$ particles/g |
| Chemical compound, drug | OptiPrep | Axis-shield | Cat# AS1114542 | |
| Chemical compound, drug | Trypan blue | Solarbio | Cat# C0040 | |
| Chemical compound, drug | FITC | Bioss | Cat# D-9801 | |
| Chemical compound, drug | Hoechst 33342 stain | Beyotime | Cat# C1028 | 100× |
| Peptide, recombinant protein | rEGFP | *Huang et al., 2021* (https://doi.org/10.3389/fimmu.2021.707770) | | recombinant plasmid, prokaryotic expression, purification |
| Peptide, recombinant protein | rCREG | *Huang et al., 2021* (https://doi.org/10.3389/fimmu.2021.707770) | | recombinant plasmid, prokaryotic expression, purification |
| Biological sample (Penaeus vannamei) | Haemolymph | Shantou local farms | | Freshly isolated from Penaeus vannamei |
| Commercial assay, kit | RNAprep Pure Micro Kit | TIANGEN | Cat#DP420 | |
| Commercial assay, kit | First Strand cDNA Synthesis Kit | Beyotime | Cat#D7168M | |
| Commercial assay, kit | 3'Reagent Kits v3.1 | 10 X Genomics | 1000268 | |
| Sequence-based reagent | *CHIT1*_F | This paper | qPCR primers | GTCGAAATTCCGGCCAAAGA |
| Sequence-based reagent | *CHIT1*_R | This paper | qPCR primers | GGCCCGTTCTTGTTTGACTT |
| Sequence-based reagent | *Lyz1*_F | This paper | qPCR primers | CAAGAACTGGGAGTGCATCG |
| Sequence-based reagent | *Lyz1*_R | This paper | qPCR primers | TCTGGAAGATGCCGTAGTCC |
| Sequence-based reagent | *NAGA*_F | This paper | qPCR primers | CTACGAGGACTACGGCAACT |
| Sequence-based reagent | *NAGA*_R | This paper | qPCR primers | CGAACTCTGGGTAGCCTTCA |
| Sequence-based reagent | *EF-1α*_F | This paper | qPCR primers | GTATTGGAACAGTGCCCGTG |
| Sequence-based reagent | *EF-1α*_R | This paper | qPCR primers | ACCAGGGACAGCCTCAGTAAG |

## Experimental organisms

Shrimp was purchased from Shantou local farms. Upon delivery, the shrimp were cultured in water tanks filled with aerated seawater at 20 °C and acclimatized for 2–3 days before the experiments. All animal-related experiments were conducted in accordance with Shantou University guidelines.

## Collection of shrimp hemocytes with different treatments

Recombinant EGFP and CREG were purified, as previously described (*Huang et al., 2021*). Sixty shrimp were divided equally into three groups. One group was left untreated and labelled as control. The other two groups were injected with rEGFP or rCREG (1 µg/g), respectively. The hemolymph was collected 8 hr post-injection from each group and mixed well. Hemolymph (1.5 mL) was loaded onto an OptiPrep (Axis-shield, NO) separation solution (1.09 g/mL) and centrifuged at 2000 rpm for 10 min at 4 °C. Circulating hemocytes were concentrated between the hemolymph and the

separation solution and carefully collected with a pipettor. The collected hemocytes were stained with 0.4% trypan blue to estimate cell viability. Next, cells with >85% viability were subjected to further scRNA-seq experiments.

## Library preparation and ScRNA sequencing

The hemocyte suspensions were loaded onto a 10 X Genomics GemCode Single-cell instrument to generate single-cell GEMs. Libraries were generated using Chromium Next GEM Single Cell 3'Reagent Kits v3.1 (10X Genomics, USA). Upon dissolution of the GEM, primers containing (i) an Illumina R1 sequence (read 1 sequencing primer), (ii) a 16 nt 10 X Barcode, (iii) a 10 nt UMI, and (iv) a poly-dT primer sequence were released and mixed with the cell lysate and Master Mix. Barcoded full-length cDNAs were then reverse-transcribed from poly adenylated mRNA and amplified using PCR to generate sufficient mass for library construction. R1 (read 1 primer sequence) was added during the GEM incubation. P5, P7, a sample index, and R2 (read 2 primer sequence) were added during library construction via end repair, A-Tailing, adaptor ligation, and PCR. Reads 1 and 2 were standard Illumina sequencing primers used for paired-end sequencing. All raw sequencing data were stored in the genome sequence archive of the Beijing Institute of Genomics, Chinese Academy of Sciences, gsa.big.ac.cn (accession nos. PRJCA006297).

## Bioinformatics analysis of single-cell RNA sequencing data

Raw BCL files were converted into FASTQ files using 10 X Genomics Cell Ranger software (version 5.0). The reads were then mapped to the shrimp genome (taxid:6689), and the reads that uniquely intersected at least 50% of an exon were considered for UMI counting. Valid barcodes were identified using the EmptyDrops method (*Lun et al., 2019*). The hemocytes by gene matrices for control, rEGFP treatment and rCREG treatment were individually imported to Seurat version 3.1.1 for the following analyses (*Butler et al., 2018*).

Cells with UMIs (≥17,000), mitochondrial genes (≥10%), ≤230 detected genes, or ≥2200 detected genes were excluded. The qualified cells were normalized via 'LogNormalize' method, which normalizes the gene expression for each cell by the total expression. The formula is as follows:

$$A\ gene\ expression\ level = log\left(1 + UMI_A/UMI_{Total} \times 10000\right)$$

The batch effect was corrected using a canonical correlation analysis (*Stuart et al., 2019*). The integrated expression matrix was then scaled and subjected to principal component analysis (PCA) for dimensional reduction. Subsequently, the significant principal components (PCs) were identified as those with a strong enrichment of low-p-value genes (*Chung and Storey, 2015*).

Seurat was used for cell clusters based on principal component analysis (PCA) scores with a subset of the data (1% by default), constructing a 'null distribution' of gene scores. A graph-based approach that calculates the distance based on previously identified PCs was implemented for cell clustering (*Chung and Storey, 2015*). Finally, a resolution of 0.2 was chosen as the clustering parameter, which identified 8 clusters. The t-SNE was then performed to visualize the data in a two-dimensional space.

Differentially expressed gene (up-regulation) analysis: The median expression patterns across all cells in each cluster were calculated to identify genes that were enriched in a specific cluster. The expression value of each gene in the given clusters was compared against the rest of the cells using the Wilcoxon rank-sum test (*Camp et al., 2017*). The significantly upregulated genes were identified using several criteria. First, genes had to be at least 1.28-fold overexpressed in the target cluster. Second, genes had to be expressed in more than 25% of the cells belonging to the target cluster. Third, the p-value had to be less than 0.05.

Cell trajectory analysis: Single-cell trajectories were analyzed using a matrix of cells and gene expression by Monocle (Version2.10.1) (*Trapnell et al., 2014*). Monocle reduced the space to one with two dimensions and ordered the cells (sigma = 0.001, lambda = NULL, param.gamma=10, tol = 0.001) (*Qiu et al., 2017*). Once the cells were ordered, we visualized the trajectory in reduced dimensional space. The trajectory had a tree-like structure, including tips and branches. Monocle was used to identify genes that were differentially expressed between the groups of cells. The key genes were identified as having a false discovery rate (FDR)<1e-5. Additionally, genes with similar trends in expression such as shared common biological functions and regulators were grouped. Finally, the Monocle

developed BEAM to analyze branch-dependent gene expression by formulating the problem as a contrast between the two negative binomial GLMs.

### Phagocytic cell labeling and sorting

Shrimp phagocytic cells were labeled as previously described (*Huang et al., 2021*). In brief, FITC labeled *Vibrio parahaemolyticus* (VP) ($2 \times 10^6$ particles/g) were injected into the shrimp. The hemocytes were collected from 30 to 40 shrimp 2 hr post-injection. Each sorting was performed on a FACSMelody cell sorter (BD Biosciences, USA). The fluorescence boundary was set based on the detection of shrimp hemocyte self-fluorescence without VP injection.

### Morphological analysis of sorted hemocyte and phagocytosis inhibition assay

Phagocytic hemocytes (R1) were collected, stained with Hoechst 33342 (Beyotime, Shanghai, China), and observed using an LSM800 confocal microscope (Zeiss, Germany). The phagocytosis inhibition assay was performed according to a previously described method with some modifications (*Kokhanyuk et al., 2021*). In brief, each shrimp was injected with either FITC-VP ($2 \times 10^6$ particles/g) or FITC-VP ($2 \times 10^6$ particles/g)+cytochalasin D (5 μM/g). The hemocytes were collected 2 hr post-injection and immediately analyzed with a BD Accuri C6 Plus Flow Cytometer (Becton Dickinson, USA). The phagocytic hemocytes were quantified based on fluorescence intensity, and the fluorescence boundary was set based on the detection of self-fluorescence of untreated hemocytes.

### Collection of sorted hemocyte RNA and proteins for RT-qPCR and immunoblot analyses

For each experiment, 50–100 k events from phagocytic hemocytes (R1) and control hemocytes (R2) were collected. Total RNA from the collected samples was purified using the RNAprep Pure Micro Kit (TIANGEN, Beijing, China) and reverse-transcribed into cDNA using a First Strand cDNA Synthesis Kit (Beyotime, Shanghai, China). qPCR was performed as previously described (*Luo et al., 2022*; *Supplementary file 10*), and the gene expression level was recorded as relative expression to *EF-1α*. This experiment was repeated five times. Total proteins from sorted hemocytes were precipitated by adding 1/100 volume of 2% sodium deoxycholate (Macklin, Shanghai, China) and 1/10 volume of 100% trichloroacetic acid (Macklin), followed by vortexing and centrifugation at 15,000×g for 15 min at 4 °C. The pellet was collected for performing SDS-PAGE and immunoblotting, as described before (*Luo et al., 2022*). This experiment was repeated thrice. The following antibodies were used: β-actin (AF5003; Beyotime, Shanghai, China), anti-NAGA (13686-T24; SinoBiological, Beijing, China), and anti-LYZ1 (bs-0816R; Bioss Antibodies, MA, USA). The polypeptide antibody against shrimp NLRP3 (aa29-42) was prepared by GenScript (Nanjing, China).

### Statistical Analyses

The data in this study are presented as the results of at least three independent experiments. Statistical analyses were performed using the GraphPad Prism 8.0. Two-tailed unpaired Student's t-tests were used to calculate the significance at $*p < 0.05$, $**p < 0.01$, and $***p < 0.001$.

## Acknowledgements

This work was supported by the National Natural Science Foundation of China (41976123 to FW), Sail Plan Program for the Introduction of Outstanding Talents of Guangdong Province of China (14600703 to FW), and 2020 Li Ka Shing Foundation Cross-Disciplinary Research Grant (2020LKSFG01E to YZ).

## Additional information

#### Competing interests

Ling Cheng, Hao Wu: are employees of Guangzhou Genedenovo Biotechnology Company Limited. The other authors declare that no competing interests exist.

## Funding

| Funder | Grant reference number | Author |
| --- | --- | --- |
| National Natural Science Foundation of China | 41976123 | Fan Wang |
| Guangdong Science and Technology Department | 14600703 | Fan Wang |
| Li Ka Shing Foundation | 2020LKSFG01E | Yueling Zhang |

The funders had no role in study design, data collection and interpretation, or the decision to submit the work for publication.

## Author contributions

Peng Yang, Validation, Investigation, Methodology, Writing – review and editing; Yaohui Chen, Validation, Investigation; Zhiqi Huang, Investigation, Methodology; Huidan Xia, Investigation; Ling Cheng, Hao Wu, Software, Formal analysis; Yueling Zhang, Funding acquisition, Writing – review and editing; Fan Wang, Conceptualization, Data curation, Supervision, Funding acquisition, Methodology, Writing - original draft, Project administration, Writing – review and editing

## Author ORCIDs

Peng Yang ⓘ http://orcid.org/0000-0002-8298-8926
Yaohui Chen ⓘ http://orcid.org/0000-0002-4044-4373
Fan Wang ⓘ http://orcid.org/0000-0002-6059-6956

## Ethics

All the animal-related experiments were in accordance with Shantou University guidelines.

## Decision letter and Author response

Decision letter https://doi.org/10.7554/eLife.80127.sa1
Author response https://doi.org/10.7554/eLife.80127.sa2

# Additional files

## Supplementary files

• Supplementary file 1. A CSV table containing specifically expressed marker genes for transitional cell (TC).

• Supplementary file 2. A CSV table containing specifically expressed marker genes for germ-like cell (GC).

• Supplementary file 3. A CSV table containing specifically expressed marker genes for prohemocyte 1 (PH1).

• Supplementary file 4. A CSV table containing specifically expressed marker genes for granulocyte 2 (GH2).

• Supplementary file 5. A CSV table containing specifically expressed marker genes for monocytic hemocyte 2 (MH2).

• Supplementary file 6. A CSV table containing specifically expressed marker genes for prohemocyte 2 (PH2).

• Supplementary file 7. A CSV table containing specifically expressed marker genes for granulocyte 1 (GH1).

• Supplementary file 8. A CSV table containing specifically expressed marker genes for monocytic hemocyte 1 (MH1).

• Supplementary file 9. A CSV table containing differentially expressed genes between monocytic hemocyte lineage and granulocyte lineage.

• Supplementary file 10. A CSV table containing a comparison between this study and other invertebrate single-cell cellular immunity studies.

• Supplementary file 11. A CSV table containing sequence alignment between shrimp marker genes in this study and its human homolog.

• MDAR checklist

## Data availability

The sequence data reported in this paper have been deposited in the Genome Sequence Archive of the Beijing Institute of Genomics, Chinese Academy of Sciences, accession no. PRJCA006297. All other data are available in this manuscript and online in the Supplementary Material.

The following dataset was generated:

| Author(s) | Year | Dataset title | Dataset URL | Database and Identifier |
|-----------|------|---------------|-------------|-------------------------|
| Wang Fan | 2022 | Single-cell RNA sequencing for shrimp (Penaeus vannamei) hemocytes treated with recombinant CREG | https://ngdc.cncb.ac.cn/bioproject/browse/PRJCA006297 | Genome Sequence Archive, PRJCA006297 |

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
