## [Editor Report]

This study provides a single cell transcriptomic atlas of shrimp hemocytes and identifies a subset of myeloid cells with markers that resemble mammalian macrophages. These novel phagocytic macrophage subset may be the target of future studies in diseased shrimp.

---

## [Decision Letter]

**Decision letter after peer review:**

Thank you for submitting your article "Myeloid cell evolution uncovered by shrimp immune cell analysis at single-cell resolution" for consideration by *eLife*. Your article has been reviewed by 3 peer reviewers, including Irene Salinas as Reviewing Editor and Reviewer #1, and the evaluation has been overseen by Carla Rothlin as the Senior Editor. The following individual involved in the review of your submission has agreed to reveal their identity: Beatriz Novoa (Reviewer #2).

Essential revisions:

1) The title needs to be revised according to the reviewers' comments.

2) Please revise the hypothesis and goal of the study in the introduction so that the experimental work actually matches the hypothesis and goals. This is very important because all three reviewers agree that the evolutionary hypothesis presented is vague and not the main point of the work.

3) It is critical to address all the experimental concerns regarding the phagocytosis assays and data presented in Figure 3 (see reviewer #1 comments).

4) Please include light microscopy or electron microscopy images of sorted phagocytes.

5) Please include in a supplementary table the % nucleotide and % amino acid identity scores between vertebrate and shrimp gene markers used in the single cell RNA-Seq dataset.

6) As brought up by all reviewers, data interpretation and discussion need to be completely reorganized. The discussion should be much more thorough and include very careful interpretation of the findings.

*Reviewer #1 (Recommendations for the authors):*

The present study provides a single cell transcriptional atlas of the white shrimp, P. vannamei, immune cells in the hemolymph, known as hemocytes. White shrimp ScRNA-Seq studies uncovered two macrophage-like populations, one of them with markers similar to mammalian macrophages. The study also shows that this new population is phagocytic and expresses proteins such as NLRP3, LYS, and NAGA. These findings redefine the current classification of shrimp immune cells which has been done using morphological approaches and via targeted qPCR studies but never using single cell transcriptomics.

Currently, the manuscript is heavily focused on the single cell dataset which should be the start to formulate hypotheses, not the main focus of the paper. The authors do a good job at identifying the MH2 subset and suggesting that they are phagocytic but further experiments are needed to substantiate this observation and increase the significance of these results.

Abstract:

1. The following sentence does not make any sense "How myeloid cells evolved from invertebrate to vertebrate is still a mystery." I suggest rephrasing to: Despite the importance of innate immunity in invertebrates, the diversity and function of myeloid cells in invertebrates are largely unknown.

2. The abstract needs to be rewritten to just say: Using single-cell RNA-Seq we uncover XXX in the white shrimp hemolymph. Our results identified a novel macrophage-like subset defined by the expression of the XXX gene MH2. MH2+ hemocytes are phagocytic and express markers that indicate some conservation with mammalian macrophages.

3. Combined, our work resolves the heterogeneity of hemocytes in the very economically important aquatic species and identifies a novel myeloid subset that is likely a critical player in the immune responses of shrimp to threatening infectious diseases affecting this industry.

Introduction:

The introduction should start by stating the importance of innate immunity in invertebrates since there is no B and T cell-based adaptive immunity. Innate immune cells in invertebrates are also very important because they have trained immunity.

After the introductory paragraph, the authors can describe what is known based on single-cell RNA-Seq from invertebrate hemocytes, basically what is already written in the current version of the manuscript from lines 66-79.

Line 80-81: I do not think the question is properly formulated. Of course, there is a common ancestor between vertebrates and invertebrates, and therefore there may be a conservation of many myeloid-like characteristics in invertebrate and vertebrate myeloid subsets. However, invertebrates may have also evolved their own subsets of myeloid cells over evolutionary time. Some may have co-opted some of the same molecules that vertebrate myeloid cells also use but many new functions may have been acquired from other genes/gene families that may have been expanded. I really beg authors to ask themselves again if this is the real question they want to answer.

Line 82: Intrigued by this question, we here use the marine invertebrate XXX to characterize immune cell subsets using scRNA-Seq. The reason to select this species is because of XXX.

Line 91: remove various.

Lines 94-99 read very strangely. If this is the end of the introduction, the authors need to frame the conceptual question and the goal of the study. For instance, if single-cell RNA-Seq has not been done before in P. vannamei please state so.

Line 204: dendritic, not dendric.

Results:

– According to Figure 1C, MH cells express not only lyz1, nlrp3 but also VEGF5. The authors do not comment on this very interesting finding. VEGF expression may confer MH cells the ability to penetrate tissues and induce angiogenesis. I do not see any VEGF5 expression data in Figure 2 so I do not know if it was differentially expressed in MH1 and MH2 populations. Can the authors please clarify? If VEGF5 is, in fact, a marker for MH2 it needs to be shown and discussed in the discussion given that VEGF is also a marker for Hofbauer cells (see my comments below). Lastly, the VEGF pathway plays a very important role in the immune response of shrimp to infection with the very important virus WSSV (doi: 10.3389/fimmu.2017.01457). Because of this, this finding is important and deserves careful examination and discussion.

– There are three unknown genes that chiefly differentiate the MH1 and MH2 clusters. These three genes are highly expressed in MH1 but not MH2 according to Figure 2B. Did the authors try to blast these three unknown genes and identify motifs/superfamilies that may suggest function? I highly suggest trying to do that and add that to the manuscript if any interesting findings emerge.

– In figure 3A: gene markers for MH1 not only point towards classical human macrophages but also Hofbauer cells, which is a unique villous macrophage population in the placenta with fetal origin. This finding is however not further commented on by the authors in the text and is something that needs to be highlighted in the manuscript. These placental macrophages also have phagocytic capabilities and therefore this finding deserves careful examination.

– Figure 3B: please show an FSC/SSC plot to show the gating strategy based on morphology prior to the FITC gating strategy.

– Interpretation of phagocytosis assays: do authors suggest that only MH2 cells are phagocytic or does the GFP+ phagocytic population include cells from other single-cell clusters?

– Phagocytosis assays: please show IF images of the in vivo assay following isolation of hemocytes and imaging of cells with ingested Vibrio GFP in the cytoplasm to confirm true phagocytosis as well as to visualize the morphology of the MH2 population.

– The morphology of the sorted phagocytic population used for qPCR and western blot studies is very important since it may show heterogeneity. It is also very important for the last figure where the authors attempt to compare the past hemocyte classification with their findings. Finally, it is important because of the Hofbauer cell villous morphology. Hofbauer cells have small nuclei and large, highly vacuolated cytoplasm so let's see if this matches with MH2 morphology.

– Phagocytosis control: please conduct ex vivo phagocytosis assays with hemocytes exposed to vibrio GFP in the presence or absence of phagocytosis inhibitors such as inhibitors of actin polymerization.

Key experiments need to be performed pertaining to increased experimental evidence to characterize the MH2 subset.

For instance, really important questions are whether this subset is found in tissues (resident macrophages), when they first appear during shrimp ontogeny, whether dietary immunostimulation or infection alters the numbers of MH2, and their transcriptional profile or even whether they display trained immunity. I do not think the authors need to answer all these questions in this one study but at least one more functional experiment needs to be done to elevate the paper's significance.

Discussion:

– The discussion is very short, lacks structure, and is poorly written. There are grammar errors in the first paragraph. For example: for animals, not animal. Please check the grammar.

– Discussions must always acknowledge the limitations of the study. For instance, here, the authors did not evaluate whether MH2 populations expand during the course of an infection/experimental challenge, which would have made the paper a lot better.

– Another limitation of the paper is whether or not MH2 cells are able to reside within tissues or they are only a circulating population. Please discuss.

– The discussion does not cover in depth any of the findings of the manuscript. For instance, there is a sentence on the NLPR3 expression, but there is no deep discussion of what it means. NLPR3 expression appears to define MH cells even at the steady state but it increases upon phagocytosis according to figure 3. Please discuss these findings. The same goes for NAGA expression, it needs to be further interpreted in the discussion.

– Line 271-273: the citations for teleost RBC being phagocytic need to be more comprehensive, many old papers had shown that not just the 2021 citation provided by the authors.

*Reviewer #2 (Recommendations for the authors):*

The authors have treated the shrimps with recombinant CREG, however, with the exception of showing that it did not induce cell differentiation, it is not mentioned in the article, nor in the abstract. Even the name of the protein has not been included.

I do not understand the relevance of fish red blood cells in the discussion.

The authors mention that they: "explained some debates in this field". This should be clarified and better explained. Maybe they forgot to include this information.

Also, the questions included in the discussion are not answered or discussed: "For example, why hyalinocytes have both proliferating activity and phagocytic activity (Soderhall, 2016). Why semi-granulocytes have phagocytic activity(M. Sun et al., 2020)."

In summary, the article is correct, but the declared general aim compared with the results is overrated.

*Reviewer #3 (Recommendations for the authors):*

I have a number of comments that require clarification before I would be willing to endorse the publication of this manuscript.

1. The written manuscript, particularly the introduction, does not align with the expectations set by the manuscript title. As it is written, this paper seems as though it is an investigation into the role of CREG in hemocyte development/differentiation. It seems as though the expectations that CREG would influence hemocyte development did not pan out, and the myeloid evolution angle became the focus. The manuscript introduction and discussion should be refocused to better guide the reader through other invertebrate studies, some of which are single-cell analyses, that contribute to our current understanding of myeloid cell evolution.

2. The methods are generally vague and at times, seem conflicting. For example, DropSeq is mentioned in the Figure 1 caption whereas 10x is mentioned elsewhere. More detail could be included for nearly all method sections.

3. Based on the information provided, primarily in the supplemental tables, it is unclear how similar the vertebrate homologs are to the shrimp genes. These vertebrate factors are used to assign a function to the shrimp transcripts, however, without a % nucleotide or amino acid identity, it is not possible to evaluate how appropriate those putative functions are. This is a cruz of the manuscript, as the conclusions that these factors can be used to define hemocyte subtypes as evolutionarily related to vertebrate myeloid cells appear to be hinged on the fact that they are functionally the same. The % nucleotide and % amino acid identity scores should be included in the supplementary tables.

4. I do not think it is sufficient to suggest that expression patterns of marker genes are strong evidence of developmental relationships between hemocyte subtypes. The authors even use the word 'probably' on line 177 when explaining this analysis. To demonstrate the differentiation relationships being proposed, the authors should sort the hemocyte subtypes using the markers identified and then experimentally show that those subtypes proposed as being 'progenitor cells' can in fact differentiate into effector subtypes.

5. The discussion is incredibly brief and speculative. Statements such as "Crustacean seems possess less phagocytic cells compared with other species, which may be due to its unique endosymbiosis with microbes in its hemolymph, which has been partially unveiled by recent studies" are unsupported and seem tangential to the stated theme of the manuscript. In other instances, the use of terms such as 'a lot' (line 171) seems sloppy and does little to inform the reader. Substantial effort should be directed at focusing the manuscript towards the intended theme.

6. Given the breadth of invertebrate species out there, it seems inappropriate to suggest that shrimp hemocytes are the only invertebrate key to unlocking myeloid cell evolution. The authors should, at the least, create a summary table of other single-cell analyses and hematopoietic hemocyte studies undertaken in invertebrates, highlighting whether there is consistency with the findings of this study in other vertebrate phyla. This type of analysis would lend itself to supporting a primarily sequence-based conclusion of myeloid cell evolution from an invertebrate origin.

[Editors’ note: further revisions were suggested prior to acceptance, as described below.]

Thank you for resubmitting your work entitled **"**Single cell RNA sequencing analysis of shrimp immune cells identifies macrophage-like phagocytes**"** for further consideration by *eLife*. Your revised article has been evaluated by Carla Rothlin (Senior Editor) and a Reviewing Editor.

The manuscript has been improved but there are some remaining issues that need to be addressed, as outlined below.

Specifically, reviewers have still identified issues with the English grammar which needs further improvement, the comparison with other invertebrate single-cell data sets beyond arthropods, and the tone down of some of the conclusions.

*Reviewer #5 (Recommendations for the authors):*

1) In the manuscript entitled: "Single cell RNA sequencing analysis of shrimp immune cells identifies macrophage-like phagocytes" by Peng Yang et al. The authors describe the characterization on a single cell level of hemocytes and immunocytes of the Shrimp Penaeus vannamei. Moreover, they find on the molecular level similarities to vertebrate myeloid lineage, specifically macrophage-like cells. They show that those cells are functionally phagocytic cells and have macrophage markers on the RNA and protein levels. Moreover, they compare their findings with the previous characterization of shrimp hemocytes.

2) This is important work since the authors use a commercially important species. The single-cell analysis is laying the base for future cellular understanding. The authors validate their sequencing data for the macrophage-like population using phagocytic function and RT-PCR and protein validation. The manuscript is well written and explained.

There are some overstatements that should be addressed. While the authors are very couscous in making homology overstatement regarding the macrophage-like cells, they are using the myeloid cell lineage as a given to exist and be homolog between vertebrates and invertebrates. Due to that please revise the second sentence in the abstract (line 19) to "innate immune cells" or something like that instead of myeloid. The same is also true for the third line of the discussion (lines 247-248). Since myeloid is a specifically defined subpopulation in vertebrates' hematopoietic system, it is not parallel to phagocytes or innate immune cells. The authors by the end of the manuscript can discuss their findings and that the myeloid lineage might evolve in arthropods already.

An additional issue related to that is to do with the first paragraph of the introduction, while the authors are trying to make a general statement of myeloid lineage in invertebrates, they give only examples from other arthropods. I think to make it a general discussion they should discuss additional works, such as the work of Rosental et al., Nature 2018, where myeloid lineage was shown in tunicates (Botryllus) on the molecular and functional levels. This way the authors can make a more general statement about the earlier emergence of the myeloid lineage.

---

## [Author Response]

Essential revisions:1) The title needs to be revised according to the reviewers' comments.

Thanks for your kindly suggestion. The new title has been focused on the major finding of this manuscript: “Single cell RNA sequencing analysis of shrimp immune cells identifies macrophage-like phagocytes”.

2) Please revise the hypothesis and goal of the study in the introduction so that the experimental work actually matches the hypothesis and goals. This is very important because all three reviewers agree that the evolutionary hypothesis presented is vague and not the main point of the work.

Thanks for all three reviewers’ valuable suggestion. We have carefully revised the introduction mainly according to reviewer1’s suggestion. In general, we have weakened the evolution hypothesis and addressed that we identified a macrophage-like phagocyte in shrimp hemolymph.

3) It is critical to address all the experimental concerns regarding the phagocytosis assays and data presented in Figure 3 (see reviewer #1 comments).

Thanks for your kindly suggestion. We carefully revised this part and added Figure 3C (confocal microscopy) and Figure 3D (phagocytosis inhibition assay) to justify that the sorted phagocytes were the cells which engulf bacteria.

4) Please include light microscopy or electron microscopy images of sorted phagocytes.

Thanks for your kindly suggestion. We have added Figure 3C (light microscopy). For electron microscopy, we have tried hard but didn’t get enough amount for TEM sample preparation. This group of phagocytes are extremely fragile and collection rate is very low even with fixation.

5) Please include in a supplementary table the % nucleotide and % amino acid identity scores between vertebrate and shrimp gene markers used in the single cell RNA-Seq dataset.

Thanks for your kindly suggestion. We have added supplementary file 11 to indicate the % nucleotide and % amino acid identity scores between vertebrate and shrimp gene markers used in this study.

6) As brought up by all reviewers, data interpretation and discussion need to be completely reorganized. The discussion should be much more thorough and include very careful interpretation of the findings.

Thanks for your kindly suggestion. We have carefully revised the whole discussion including comprehensive understanding of the findings.

Reviewer #1 (Recommendations for the authors):The present study provides a single cell transcriptional atlas of the white shrimp, P. vannamei, immune cells in the hemolymph, known as hemocytes. White shrimp ScRNA-Seq studies uncovered two macrophage-like populations, one of them with markers similar to mammalian macrophages. The study also shows that this new population is phagocytic and expresses proteins such as NLRP3, LYS, and NAGA. These findings redefine the current classification of shrimp immune cells which has been done using morphological approaches and via targeted qPCR studies but never using single cell transcriptomics.

Thanks for your kindly comment. We have uncovered one macrophage-like population and another population is granulocyte.

Currently, the manuscript is heavily focused on the single cell dataset which should be the start to formulate hypotheses, not the main focus of the paper. The authors do a good job at identifying the MH2 subset and suggesting that they are phagocytic but further experiments are needed to substantiate this observation and increase the significance of these results.

Thanks for your kindly comments. We have added confocal microscopy image (Figure 3C) and phagocytosis inhibition assay (Figure 3D) for verification of these sorted phagocytes.

Abstract:1. The following sentence does not make any sense "How myeloid cells evolved from invertebrate to vertebrate is still a mystery." I suggest rephrasing to: Despite the importance of innate immunity in invertebrates, the diversity and function of myeloid cells in invertebrates are largely unknown.

Thanks for your valuable comments, we have followed your suggestion and revised this part in abstract (Line18-19).

2. The abstract needs to be rewritten to just say: Using single-cell RNA-Seq we uncover XXX in the white shrimp hemolymph. Our results identified a novel macrophage-like subset defined by the expression of the XXX gene MH2. MH2+ hemocytes are phagocytic and express markers that indicate some conservation with mammalian macrophages.

Thanks for your valuable comments, we have followed your suggestion and revised this part in abstract (Line19-25).

3. Combined, our work resolves the heterogeneity of hemocytes in the very economically important aquatic species and identifies a novel myeloid subset that is likely a critical player in the immune responses of shrimp to threatening infectious diseases affecting this industry.

Thanks for your valuable comments, we have followed your suggestion and revised this part in abstract (Line25-28).

Introduction:The introduction should start by stating the importance of innate immunity in invertebrates since there is no B and T cell-based adaptive immunity. Innate immune cells in invertebrates are also very important because they have trained immunity.After the introductory paragraph, the authors can describe what is known based on single-cell RNA-Seq from invertebrate hemocytes, basically what is already written in the current version of the manuscript from lines 66-79.Line 80-81: I do not think the question is properly formulated. Of course, there is a common ancestor between vertebrates and invertebrates, and therefore there may be a conservation of many myeloid-like characteristics in invertebrate and vertebrate myeloid subsets. However, invertebrates may have also evolved their own subsets of myeloid cells over evolutionary time. Some may have co-opted some of the same molecules that vertebrate myeloid cells also use but many new functions may have been acquired from other genes/gene families that may have been expanded. I really beg authors to ask themselves again if this is the real question they want to answer.

Thanks for your kindly suggestion. We recognize that the evolution is so complex and my study in this manuscript could not lead to such a conclusion. Thus, we have deleted line80-81.

Line 82: Intrigued by this question, we here use the marine invertebrate XXX to characterize immune cell subsets using scRNA-Seq. The reason to select this species is because of XXX.Line 91: remove various.Lines 94-99 read very strangely. If this is the end of the introduction, the authors need to frame the conceptual question and the goal of the study. For instance, if single-cell RNA-Seq has not been done before in P. vannamei please state so.Line 204: dendritic, not dendric.

Thanks for your valuable comments, we have revised this part as you suggestion.

Results:– According to Figure 1C, MH cells express not only lyz1, nlrp3 but also VEGF5. The authors do not comment on this very interesting finding. VEGF expression may confer MH cells the ability to penetrate tissues and induce angiogenesis. I do not see any VEGF5 expression data in Figure 2 so I do not know if it was differentially expressed in MH1 and MH2 populations. Can the authors please clarify? If VEGF5 is, in fact, a marker for MH2 it needs to be shown and discussed in the discussion given that VEGF is also a marker for Hofbauer cells (see my comments below). Lastly, the VEGF pathway plays a very important role in the immune response of shrimp to infection with the very important virus WSSV (doi: 10.3389/fimmu.2017.01457). Because of this, this finding is important and deserves careful examination and discussion.

Thanks for your kindly reminder. Figure 1C shows that MH cells express VEGF3. Below are the reasons that why we don’t talk about VEGF:

1. Currently, five VEGF subtypes (VEGF1-5) have been identified in shrimp (doi: 10.1016/j.fsi.2015.10.026 doi: 10.1016/j.dci.2016.05.020 doi: 10.1016/j.fsi.2018.10.019). However, shrimp holds an open circulation system with partial blood vessel. Until now it is not quite sure that whether angiogenesis in adult is important or not.

2. Besides VEGF3, other VEGFs could express almost all tissues in shrimp (doi: 10.1016/j.fsi.2015.10.026 doi: 10.1016/j.dci.2016.05.020 doi: 10.1016/j.fsi.2018.10.019) although different subtypes may have some tissue preference. People don’t know what’s the difference for all these VEGF subtypes in shrimp. Here, we used bubble chart to characterize the expression of VEGF3 in different types of blood cells of shrimp (see Author response image 1).

**Author response image 1. sa2fig1:** This Figure indicates that VEGF3 has no significant differentially expression in MH1 and MH2 populations. Thus, we don’t list this gene in Figure 2B and give a discussion for this marker. Moreover, it is difficult to address this point because VEGFs in shrimp have five subtypes. All five subtypes play some roles during WSSV infection (doi: 10.1016/j.fsi.2015.10.026 doi: 10.1016/j.dci.2016.05.020 doi: 10.1016/j.fsi.2018.10.019). Only VEGF3 was highly expressed in MH subtype. This system is too complicated and far away from being clarified at this moment. Thus, it is very difficult to discussion this part now.

– There are three unknown genes that chiefly differentiate the MH1 and MH2 clusters. These three genes are highly expressed in MH1 but not MH2 according to Figure 2B. Did the authors try to blast these three unknown genes and identify motifs/superfamilies that may suggest function? I highly suggest trying to do that and add that to the manuscript if any interesting findings emerge.

Thanks for your valuable suggestion. Author response image 2 is the blast results in Pubmed.

**Author response image 2. sa2fig2:** This Image represents the top three genes are quite similar and only conserved in shrimp. The blast results don’t show much information about these proteins’ function.

– In figure 3A: gene markers for MH1 not only point towards classical human macrophages but also Hofbauer cells, which is a unique villous macrophage population in the placenta with fetal origin. This finding is however not further commented on by the authors in the text and is something that needs to be highlighted in the manuscript. These placental macrophages also have phagocytic capabilities and therefore this finding deserves careful examination.

Thanks for your valuable suggestion. we have added some discussion about this point (Line189-Line190). Currently very few information about invertebrate phagocytes has been studied. Further study is needed after preparation of monoclonal antibodies recognizing this subtype, which is under the way in my lab.

– Figure 3B: please show an FSC/SSC plot to show the gating strategy based on morphology prior to the FITC gating strategy.

Thanks for your valuable comments. We have sorted out a typical FSC/SSC plot for the gating strategy (Author response image 3).

**Author response image 3. sa2fig3:** These images show that setting a suitable gating strategy to excluded interference factors to visualize FITC.

– Interpretation of phagocytosis assays: do authors suggest that only MH2 cells are phagocytic or does the GFP+ phagocytic population include cells from other single-cell clusters?

Thanks for your question, we cannot say only MH2 cells are phagocytic at this moment. In this study MH2 cell is 1.9% of total hemocytes. According to our previous study and other group study (doi: 10.3389/fimmu.2021.707770; doi: 10.4049/jimmunol.1900156.), shrimp phagocyte rate is around 1% to 10% of total hemocytes with huge individual variation. GFP+ phagocytic population may include cells from other single-cell clusters.

– Phagocytosis assays: please show IF images of the in vivo assay following isolation of hemocytes and imaging of cells with ingested Vibrio GFP in the cytoplasm to confirm true phagocytosis as well as to visualize the morphology of the MH2 population.

Thanks for your kindly suggestion. We have added the confocal image for isolated phagocytes which ingested FITC-VP (Figure 3C).

– The morphology of the sorted phagocytic population used for qPCR and western blot studies is very important since it may show heterogeneity. It is also very important for the last figure where the authors attempt to compare the past hemocyte classification with their findings. Finally, it is important because of the Hofbauer cell villous morphology. Hofbauer cells have small nuclei and large, highly vacuolated cytoplasm so let's see if this matches with MH2 morphology.

Thanks for your kindly comment. We have added confocal microscopy image in this manuscript (Figure 3C). As shown in Figure 3C, the shrimp phagocytes have round nuclei with highly vacuolated cytoplasm which is a typical macrophage morphology (Line 198-200).

– Phagocytosis control: please conduct ex vivo phagocytosis assays with hemocytes exposed to vibrio GFP in the presence or absence of phagocytosis inhibitors such as inhibitors of actin polymerization.Key experiments need to be performed pertaining to increased experimental evidence to characterize the MH2 subset.For instance, really important questions are whether this subset is found in tissues (resident macrophages), when they first appear during shrimp ontogeny, whether dietary immunostimulation or infection alters the numbers of MH2, and their transcriptional profile or even whether they display trained immunity. I do not think the authors need to answer all these questions in this one study but at least one more functional experiment needs to be done to elevate the paper's significance.

Thanks for your kindly suggestion. We have added both confocal image (Figure 3C) and phagocytosis inhibition assay (Figure 3D) to characterize the phagocytes which ingested FITC-VP. For the question about resident macrophages, how it changes with immunostimulation or infection. All these important questions required high quality antibodies for certain MH2 marker genes, which we are working hard to screen out.

Discussion:– The discussion is very short, lacks structure, and is poorly written. There are grammar errors in the first paragraph. For example: for animals, not animal. Please check the grammar.

Thanks for your kindly comment. We have rewritten the discussion.

– Discussions must always acknowledge the limitations of the study. For instance, here, the authors did not evaluate whether MH2 populations expand during the course of an infection/experimental challenge, which would have made the paper a lot better.

Thanks for your kindly comment. We are working hard to screen monoclonal antibodies for MH2 surface marker and examine these issues in future studies.

– Another limitation of the paper is whether or not MH2 cells are able to reside within tissues or they are only a circulating population. Please discuss.

Thanks for your kindly suggestion. We have added line 279-284 to discuss this issue.

– The discussion does not cover in depth any of the findings of the manuscript. For instance, there is a sentence on the NLPR3 expression, but there is no deep discussion of what it means. NLPR3 expression appears to define MH cells even at the steady state but it increases upon phagocytosis according to figure 3. Please discuss these findings. The same goes for NAGA expression, it needs to be further interpreted in the discussion.

Thanks for your kindly suggestion. Because we cannot isolate MH2 subset at this moment, whether Nlrp3 expression is upregulated upon phagocytosis is not clear. We have added line256 -line 267 to further discuss this finding.

– Line 271-273: the citations for teleost RBC being phagocytic need to be more comprehensive, many old papers had shown that not just the 2021 citation provided by the authors.

Thanks for your kindly suggestion. We have rewritten this part and added several references.

Reviewer #2 (Recommendations for the authors):The authors have treated the shrimps with recombinant CREG, however, with the exception of showing that it did not induce cell differentiation, it is not mentioned in the article, nor in the abstract. Even the name of the protein has not been included.

Thanks for your kindly suggestion. Indeed, this experiment was designed originally to explore shrimp plasma CREG function. However, recombinant CREG treatment doesn’t induce hemocyte differentiation and just activate the hemocytes in general which we have mentioned in line75-77 and line86-90. For this reason, we focused on identification of novel shrimp hemocyte subtype in this study.

I do not understand the relevance of fish red blood cells in the discussion.

Thanks for your kindly comment. We have rewritten the discussion of this manuscript and put focus on identification of a novel macrophage-like phagocyte in shrimp hemolymph. Thus, we tried to compare different phagocytic cells from various species and discuss their evolution(line273-284).

The authors mention that they: "explained some debates in this field". This should be clarified and better explained. Maybe they forgot to include this information.

Thanks for your kindly suggestion. We have rewritten this part and explained the debates about hyalinocytes and semi-granulocytes (Line287-300).

Also, the questions included in the discussion are not answered or discussed: "For example, why hyalinocytes have both proliferating activity and phagocytic activity (Soderhall, 2016). Why semi-granulocytes have phagocytic activity (M. Sun et al., 2020)."

Thanks for your kindly comments. I have deleted these two sentences and rewritten this part.

In summary, the article is correct, but the declared general aim compared with the results is overrated.

Thanks for your kindly suggestion. We have realized this problem and rewritten the whole manuscript in which we have shifted the manuscript focus from myeloid cell evolution to identification of an invertebrate macrophage-like phagocyte.

Reviewer #3 (Recommendations for the authors):I have a number of comments that require clarification before I would be willing to endorse the publication of this manuscript.1. The written manuscript, particularly the introduction, does not align with the expectations set by the manuscript title. As it is written, this paper seems as though it is an investigation into the role of CREG in hemocyte development/differentiation. It seems as though the expectations that CREG would influence hemocyte development did not pan out, and the myeloid evolution angle became the focus. The manuscript introduction and discussion should be refocused to better guide the reader through other invertebrate studies, some of which are single-cell analyses, that contribute to our current understanding of myeloid cell evolution.

Thanks for your kindly comment. Yes, we designed this experiment to explore CREG differentiation function at the beginning. However, CREG doesn’t show such a function so that we put focus on shrimp hemocyte subtype identification in this study. I have carefully rewritten the introduction and Discussion section to address identification of a novel microphage-like phagocyte instead of myeloid cell evolution which is too broad for this study.

2. The methods are generally vague and at times, seem conflicting. For example, DropSeq is mentioned in the Figure 1 caption whereas 10x is mentioned elsewhere. More detail could be included for nearly all method sections.

Thanks for your kindly comment. I have carefully revised the method sections which was labelled with red (Line334-345, Line364-389) and deleted the DropSeq in Figure 1 caption. The study was completely performed in 10×Genomics platform.

3. Based on the information provided, primarily in the supplemental tables, it is unclear how similar the vertebrate homologs are to the shrimp genes. These vertebrate factors are used to assign a function to the shrimp transcripts, however, without a % nucleotide or amino acid identity, it is not possible to evaluate how appropriate those putative functions are. This is a cruz of the manuscript, as the conclusions that these factors can be used to define hemocyte subtypes as evolutionarily related to vertebrate myeloid cells appear to be hinged on the fact that they are functionally the same. The % nucleotide and % amino acid identity scores should be included in the supplementary tables.

Thanks for your kindly suggestion. We have added supplementary file 11 to address this issue.

4. I do not think it is sufficient to suggest that expression patterns of marker genes are strong evidence of developmental relationships between hemocyte subtypes. The authors even use the word 'probably' on line 177 when explaining this analysis. To demonstrate the differentiation relationships being proposed, the authors should sort the hemocyte subtypes using the markers identified and then experimentally show that those subtypes proposed as being 'progenitor cells' can in fact differentiate into effector subtypes.

Thanks for your kindly suggestion. I recognized that I have over-interpreted the data shown in this manuscript. Thus, I have carefully rewritten the whole manuscript and shifted my major claim from myeloid cell evolution to identification of a novel invertebrate macrophage-like phagocytes. We are trying hard to screen the markers’ monoclonal antibodies for major subtypes listed in this study.

5. The discussion is incredibly brief and speculative. Statements such as "Crustacean seems possess less phagocytic cells compared with other species, which may be due to its unique endosymbiosis with microbes in its hemolymph, which has been partially unveiled by recent studies" are unsupported and seem tangential to the stated theme of the manuscript. In other instances, the use of terms such as 'a lot' (line 171) seems sloppy and does little to inform the reader. Substantial effort should be directed at focusing the manuscript towards the intended theme.

Thanks for your kindly comment. we have deleted the sentence” Crustacean seems possess less phagocytic cells compared with other species, which may be due to its unique endosymbiosis with microbes in its hemolymph, which has been partially unveiled by recent studies”. Moreover, we have changed “a lot of” to “some” (line147). In general, we have carefully rewritten the whole manuscript.

6. Given the breadth of invertebrate species out there, it seems inappropriate to suggest that shrimp hemocytes are the only invertebrate key to unlocking myeloid cell evolution. The authors should, at the least, create a summary table of other single-cell analyses and hematopoietic hemocyte studies undertaken in invertebrates, highlighting whether there is consistency with the findings of this study in other vertebrate phyla. This type of analysis would lend itself to supporting a primarily sequence-based conclusion of myeloid cell evolution from an invertebrate origin.

Thanks for your kindly suggestion. I have added Supplementary file 10 to compare our study with other invertebrate cellular immunity studies.

[Editors’ note: what follows is the authors’ response to the second round of review.]

Reviewer #5 (Recommendations for the authors):1) In the manuscript entitled: "Single cell RNA sequencing analysis of shrimp immune cells identifies macrophage-like phagocytes" by Peng Yang et al. The authors describe the characterization on a single cell level of hemocytes and immunocytes of the Shrimp Penaeus vannamei. Moreover, they find on the molecular level similarities to vertebrate myeloid lineage, specifically macrophage-like cells. They show that those cells are functionally phagocytic cells and have macrophage markers on the RNA and protein levels. Moreover, they compare their findings with the previous characterization of shrimp hemocytes.2) This is important work since the authors use a commercially important species. The single-cell analysis is laying the base for future cellular understanding. The authors validate their sequencing data for the macrophage-like population using phagocytic function and RT-PCR and protein validation. The manuscript is well written and explained.There are some overstatements that should be addressed. While the authors are very couscous in making homology overstatement regarding the macrophage-like cells, they are using the myeloid cell lineage as a given to exist and be homolog between vertebrates and invertebrates. Due to that please revise the second sentence in the abstract (line 19th) to "innate immune cells" or something like that instead of myeloid. The same is also true for the third line of the discussion (lines 247-248). Since myeloid is a specifically defined subpopulation in vertebrates' hematopoietic system, it is not parallel to phagocytes or innate immune cells. The authors by the end of the manuscript can discuss their findings and that the myeloid lineage might evolve in arthropods already.

Thank you for this suggestion, I have replaced “myeloid cell” with “innate immune cell” for invertebrate.

An additional issue related to that is to do with the first paragraph of the introduction, while the authors are trying to make a general statement of myeloid lineage in invertebrates, they give only examples from other arthropods. I think to make it a general discussion they should discuss additional works, such as the work of Rosental et al., Nature 2018, where myeloid lineage was shown in tunicates (Botryllus) on the molecular and functional levels. This way the authors can make a more general statement about the earlier emergence of the myeloid lineage.

Thank you for this suggestion, I have weakened the myeloid lineage claim for invertebrate and replace them with “innate immune cell”. In addition, I have mentioned the work of Rosental et al., Nature 2018 in line 39.